# Surprise-Based Intrinsic Motivation for Deep Reinforcement Learning

**Joshua Achiam & Shankar Sastry**
Department of Electrical Engineering and Computer Science
UC Berkeley
jachiam@berkeley.edu, sastry@coe.berkeley.edu

## Abstract

Exploration in complex domains is a key challenge in reinforcement learning, especially for tasks with very sparse rewards. Recent successes in deep reinforcement learning have been achieved mostly using simple heuristic exploration strategies such as $\epsilon$-greedy action selection or Gaussian control noise, but there are many tasks where these methods are insufficient to make any learning progress. Here, we consider more complex heuristics: efficient and scalable exploration strategies that maximize a notion of an agent's surprise about its experiences via intrinsic motivation. We propose to learn a model of the MDP transition probabilities concurrently with the policy, and to form intrinsic rewards that approximate the KL-divergence of the true transition probabilities from the learned model. One of our approximations results in using surprisal as intrinsic motivation, while the other gives the $k$-step learning progress. We show that our incentives enable agents to succeed in a wide range of environments with high-dimensional state spaces and very sparse rewards, including continuous control tasks and games in the Atari RAM domain, outperforming several other heuristic exploration techniques.

## 1 Introduction

A reinforcement learning agent uses experiences obtained from interacting with an unknown environment to learn behavior that maximizes a reward signal. The optimality of the learned behavior is strongly dependent on how the agent approaches the exploration/exploitation trade-off in that environment. If it explores poorly or too little, it may never find rewards from which to learn, and its behavior will always remain suboptimal; if it does find rewards but exploits them too intensely, it may wind up prematurely converging to suboptimal behaviors, and fail to discover more rewarding opportunities. Although substantial theoretical work has been done on optimal exploration strategies for environments with finite state and action spaces, we are here concerned with problems that have continuous state and/or action spaces, where algorithms with theoretical guarantees admit no obvious generalization or are prohibitively impractical to implement.

Simple heuristic methods of exploring such as $\epsilon$-greedy action selection and Gaussian control noise have been successful on a wide range of tasks, but are inadequate when rewards are especially sparse. For example, the Deep Q-Network approach of Mnih et al. [13] used $\epsilon$-greedy exploration in training deep neural networks to play Atari games directly from raw pixels. On many games, the algorithm resulted in superhuman play; however, on games like Montezuma's Revenge, where rewards are extremely sparse, DQN (and its variants [25], [26], [15], [12]) with $\epsilon$-greedy exploration failed to achieve scores even at the level of a novice human. Similarly, in benchmarking deep reinforcement learning for continuous control, Duan et al.[5] found that policy optimization algorithms that explored by acting according to the current stochastic policy, including REINFORCE and Trust Region Policy Optimization (TRPO), could succeed across a diverse slate of simulated robotics control tasks with well-defined, non-sparse reward signals (like rewards proportional to the forward velocity of the robot). Yet, when tested in environments with sparse rewards—where the agent would only be able to attain rewards after first figuring out complex motion primitives *without reinforcement*—every algorithm failed to attain scores better than random agents. The failure modes in all of these cases pertained to the nature of the exploration: the agents encountered reward signals so infrequently that they were never able to learn reward-seeking behavior.

One approach to encourage better exploration is via intrinsic motivation, where an agent has a task-independent, often information-theoretic intrinsic reward function which it seeks to maximize in addition to the reward from the environment. Examples of intrinsic motivation include empowerment, where the agent enjoys the level of control it has about its future; surprise, where the agent is excited to see outcomes that run contrary to its understanding of the world; and novelty, where the agent is excited to see new states (which is tightly connected to surprise, as shown in [2]). For in-depth reviews of the different types of intrinsic motivation, we direct the reader to [1] and [17].

Recently, several applications of intrinsic motivation to the deep reinforcement learning setting (such as [2], [7], [22]) have found promising success. In this work, we build on that success by exploring scalable measures of surprise for intrinsic motivation in deep reinforcement learning. We formulate surprise as the KL-divergence of the true transition probability distribution from a transition model which is learned concurrently with the policy, and consider two approximations to this divergence which are easy to compute in practice. One of these approximations results in using the surprisal of a transition as an intrinsic reward; the other results in using a measure of learning progress which is closer to a Bayesian concept of surprise. Our contributions are as follows:

1. we investigate surprisal and learning progress as intrinsic rewards across a wide range of environments in the deep reinforcement learning setting, and demonstrate empirically that the incentives (especially surprisal) result in efficient exploration,

2. we evaluate the difficulty of the slate of sparse reward continuous control tasks introduced by Houthooft et al. [7] to benchmark exploration incentives, and introduce a new task to complement the slate,

3. and we present an efficient method for learning the dynamics model (transition probabilities) concurrently with a policy.

We distinguish our work from prior work in a number of implementation details: unlike Bellemare et al. [2], we learn a transition model as opposed to a state-action occupancy density; unlike Stadie et al. [22], our formulation naturally encompasses environments with stochastic dynamics; unlike Houthooft et al. [7], we avoid the overhead of maintaining a distribution over possible dynamics models, and learn a single deep dynamics model.

In our empirical evaluations, we compare the performance of our proposed intrinsic rewards with other heuristic intrinsic reward schemes and to recent results from the literature. In particular, we compare to Variational Information Maximizing Exploration (VIME) [7], a method which approximately maximizes Bayesian surprise and currently achieves state-of-the-art performance on continuous control with sparse rewards. We show that our incentives can perform on the level of VIME at a lower computational cost.

## 2 PRELIMINARIES

We begin by introducing notation which we will use throughout the paper. A Markov decision process (MDP) is a tuple, $(S, A, R, P, \mu)$, where $S$ is the set of states, $A$ is the set of actions, $R : S \times A \times S \to \mathbb{R}$ is the reward function, $P : S \times A \times S \to [0, 1]$ is the transition probability function (where $P(s'|s, a)$ is the probability of transitioning to state $s'$ given that the previous state was $s$ and the agent took action $a$ in $s$), and $\mu : S \to [0, 1]$ is the starting state distribution. A policy $\pi : S \times A \to [0, 1]$ is a distribution over actions per state, with $\pi(a|s)$ the probability of selecting $a$ in state $s$. We aim to select a policy $\pi$ which maximizes a performance measure, $L(\pi)$, which usually takes the form of expected finite-horizon total return (sum of rewards in a fixed time period), or expected infinite-horizon discounted total return (discounted sum of all rewards forever). In this paper, we use the finite-horizon total return formulation.

## 3 SURPRISE INCENTIVES

To train an agent with surprise-based exploration, we alternate between making an update step to a dynamics model (an approximator of the MDP's transition probability function), and making a policy update step that maximizes a trade-off between policy performance and a surprise measure.

The dynamics model step makes progress on the optimization problem

$$\min_{\phi} -\frac{1}{|D|} \sum_{(s,a,s') \in D} \log P_\phi(s'|s,a) + \alpha f(\phi), \tag{1}$$

where $D$ is is a dataset of transition tuples from the environment, $P_\phi$ is the model we are learning, $f$ is a regularization function, and $\alpha > 0$ is a regularization trade-off coefficient. The policy update step makes progress on an approximation to the optimization problem

$$\max_{\pi} L(\pi) + \eta \, \underset{s,a \sim \pi}{E} \left[ D_{KL}(P||P_\phi)[s,a] \right], \tag{2}$$

where $\eta > 0$ is an explore-exploit trade-off coefficient. The exploration incentive in (2), which we select to be the on-policy average KL-divergence of $P_\phi$ from $P$, is intended to capture the agent's surprise about its experience. The dynamics model $P_\phi$ should only be close to $P$ on regions of the transition state space that the agent has already visited (because those transitions will appear in $D$ and thus the model will be fit to them), and as a result, the KL divergence of $P_\phi$ and $P$ will be higher in unfamiliar places. Essentially, this exploits the generalization in the model to encourage the agent to go where it has not gone before. The surprise incentive in (2) gives the net effect of performing a reward shaping of the form

$$r'(s,a,s') = r(s,a,s') + \eta \left( \log P(s'|s,a) - \log P_\phi(s'|s,a) \right), \tag{3}$$

where $r(s,a,s')$ is the original reward and $r'(s,a,s')$ is the transformed reward, so ideally we could solve (2) by applying any reinforcement learning algorithm with these reshaped rewards. In practice, we cannot directly implement this reward reshaping because $P$ is unknown. Instead, we consider two ways of finding an approximate solution to (2).

In one method, we approximate the KL-divergence by the cross-entropy, which is reasonable when $H(P)$ is finite (and small) and $P_\phi$ is sufficiently far from $P$[1]; that is, denoting the cross-entropy by $H(P, P_\phi)[s,a] \doteq E_{s' \sim P(\cdot|s,a)}[-\log P_\phi(s'|s,a)]$, we assume

$$\begin{aligned} D_{KL}(P||P_\phi)[s,a] &= H(P, P_\phi)[s,a] - H(P)[s,a] \\ &\approx H(P, P_\phi)[s,a]. \end{aligned} \tag{4}$$

This approximation results in a reward shaping of the form

$$r'(s,a,s') = r(s,a,s') - \eta \log P_\phi(s'|s,a); \tag{5}$$

here, the intrinsic reward is the surprisal of $s'$ given the model $P_\phi$ and the context $(s,a)$.

In the other method, we maximize a lower bound on the objective in (2) by lower bounding the surprise term:

$$\begin{aligned} D_{KL}(P||P_\phi)[s,a] &= D_{KL}(P||P_{\phi'})[s,a] + \underset{s' \sim P}{E} \left[ \log \frac{P_{\phi'}(s'|s,a)}{P_\phi(s'|s,a)} \right] \\ &\geq \underset{s' \sim P}{E} \left[ \log \frac{P_{\phi'}(s'|s,a)}{P_\phi(s'|s,a)} \right]. \end{aligned} \tag{6}$$

The bound (6) results in a reward shaping of the form

$$r'(s,a,s') = r(s,a,s') + \eta \left( \log P_{\phi'}(s'|s,a) - \log P_\phi(s'|s,a) \right), \tag{7}$$

which requires a choice of $\phi'$. From (6), we can see that the bound becomes tighter by minimizing $D_{KL}(P||P_{\phi'})$. As a result, we choose $\phi'$ to be the parameters of the dynamics model after $k$ updates based on (1), and $\phi$ to be the parameters from before the updates. Thus, at iteration $t$, the reshaped rewards are

$$r'(s,a,s') = r(s,a,s') + \eta \left( \log P_{\phi_t}(s'|s,a) - \log P_{\phi_{t-k}}(s'|s,a) \right); \tag{8}$$

here, the intrinsic reward is the $k$-step learning progress at $(s,a,s')$. It also bears a resemblance to Bayesian surprise; we expand on this similarity in the next section.

In our experiments, we investigate both the surprisal bonus (5) and the $k$-step learning progress bonus (8) (with varying values of $k$).

---

[1]On the other hand, if $H(P)[s,a]$ is non-finite everywhere—for instance if the MDP has continuous states and deterministic transitions—then as long as it has the same sign everywhere, $E_{s,a \sim \pi}[H(P)[s,a]]$ is a constant with respect to $\pi$ and we can drop it from the optimization problem anyway.

## 3.1 DISCUSSION

Ideally, we would like the intrinsic rewards to vanish in the limit as $P_\phi \to P$, because in this case, the agent should have sufficiently explored the state space, and should primarily learn from extrinsic rewards. For the proposed intrinsic reward in (5), this is not the case, and it may result in poor performance in that limit. The thinking goes that when $P_\phi = P$, the agent will be incentivized to seek out states with the noisiest transitions. However, we argue that this may not be an issue, because the intrinsic motivation seems mostly useful long before the dynamics model is fully learned. As long as the agent is able to find the extrinsic rewards before the intrinsic reward is just the entropy in $P$, the pathological noise-seeking behavior should not happen. On the other hand, the intrinsic reward in (8) should not suffer from this pathology, because in the limit, as the dynamics model converges, we should have $P_{\phi_t} \approx P_{\phi_{t-k}}$. Then the intrinsic reward will vanish as desired.

Next, we relate (8) to Bayesian surprise. The Bayesian surprise associated with a transition is the reduction in uncertainty over possibly dynamics models from observing it ([1],[8]):

$$D_{KL}\left(P(\phi|h_t, a_t, s_{t+1})||P(\phi|h_t)\right).$$

Here, $P(\phi|h_t)$ is meant to represent a distribution over possible dynamics models parametrized by $\phi$ given the preceding history of observed states and actions $h_t$ (so $h_t$ includes $s_t$), and $P(\phi|h_t, a_t, s_{t+1})$ is the posterior distribution over dynamics models after observing $(a_t, s_{t+1})$. By Bayes' rule, the dynamics prior and posterior are related to the model-based transition probabilities by

$$P(\phi|h_t, a_t, s_{t+1}) = \frac{P(\phi|h_t)P(s_{t+1}|h_t, a_t, \phi)}{\mathrm{E}_{\phi \sim P(\cdot|h_t)}\left[P(s_{t+1}|h_t, a_t, \phi)\right]},$$

so the Bayesian surprise can be expressed as

$$\mathop{\mathrm{E}}_{\phi \sim P_{t+1}}\left[\log P(s_{t+1}|h_t, a_t, \phi)\right] - \log \mathop{\mathrm{E}}_{\phi \sim P_t}\left[P(s_{t+1}|h_t, a_t, \phi)\right], \tag{9}$$

where $P_{t+1} = P(\cdot|h_t, a_t, s_{t+1})$ is the posterior and $P_t = P(\cdot|h_t)$ is the prior. In this form, the resemblance between (9) and (8) is clarified. Although the update from $\phi_{t-k}$ to $\phi_t$ is not Bayesian—and is performed in batch, instead of per transition sample—we can imagine (8) might contain similar information to (9).

## 3.2 IMPLEMENTATION DETAILS

Our implementation uses $L_2$ regularization in the dynamics model fitting, and we impose an additional constraint to keep model iterates close in the KL-divergence sense. Denoting the average divergence as

$$\bar{D}_{KL}(P_{\phi'}||P_\phi) = \frac{1}{|D|}\sum_{(s,a)\in D} D_{KL}(P_{\phi'}||P_\phi)[s,a], \tag{10}$$

our dynamics model update is

$$\phi_{i+1} = \arg\min_\phi -\frac{1}{|D|}\sum_{(s,a,s')\in D}\log P_\phi(s'|s,a) + \alpha\|\phi\|_2^2 \;:\; \bar{D}_{KL}(P_\phi||P_{\phi_i}) \leq \kappa. \tag{11}$$

The constraint value $\kappa$ is a hyper-parameter of the algorithm. We solve this optimization problem approximately using a single second-order step with a line search, as described by [20]; full details are given in supplementary material. $D$ is a FIFO replay memory, and at each iteration, instead of using the entirety of $D$ for the update step we sub-sample a batch $d \subset D$. Also, similarly to [7], we adjust the bonus coefficient $\eta$ at each iteration, to keep the average bonus magnitude upper-bounded (and usually fixed). Let $\eta_0$ denote the desired average bonus, and $r_+(s, a, s')$ denote the intrinsic reward; then, at each iteration, we set

$$\eta = \frac{\eta_0}{\max\left(1, \frac{1}{|B|}\left|\sum_{(s,a,s')\in B} r_+(s, a, s')\right|\right)},$$

where $B$ is the batch of data used for the policy update step. This normalization improves the stability of the algorithm by keeping the scale of the bonuses fixed with respect to the scale of the extrinsic rewards. Also, in environments where the agent can die, we avoid the possibility of the intrinsic rewards becoming a living cost by translating all bonuses so that the mean is nonnegative. The basic outline of the algorithm is given as Algorithm 1. In all experiments, we use fully-factored Gaussian distributions for the dynamics models, where the means and variances are the outputs of neural networks.

---
**Algorithm 1** Reinforcement Learning with Surprise Incentive

---
 **Input:** Initial policy $\pi_0$, dynamics model $P_{\phi_0}$
 **repeat**
 collect rollouts on current policy $\pi_i$
 add rollout $(s, a, s')$ tuples to replay memory $D$
 compute reshaped rewards using (5) or (8) with dynamics model $P_{\phi_i}$
 normalize $\eta$ by the average intrinsic reward of the current batch of data
 update policy to $\pi_{i+1}$ using any RL algorithm with the reshaped rewards
 update the dynamics model to $P_{\phi_{i+1}}$ according to (11)
 **until** training is completed

---

## 4 EXPERIMENTS

We evaluate our proposed surprise incentives on a wide range of benchmarks that are challenging for naive exploration methods, including continuous control and discrete control tasks. Our continuous control tasks include the slate of sparse reward tasks introduced by Houthooft et al. [7]: sparse MountainCar, sparse CartPoleSwingup, and sparse HalfCheetah, as well as a new sparse reward task that we introduce here: sparse Swimmer. (We refer to these environments with the prefix 'sparse' to differentiate them from other versions which appear in the literature, where agents receive non-sparse reward signals.) Additionally, we evaluate performance on a highly-challenging hierarchical sparse reward task introduced by Duan et al [5], SwimmerGather. The discrete action tasks are several games from the Atari RAM domain of the OpenAI Gym [4]: Pong, BankHeist, Freeway, and Venture.

Environments with deterministic and stochastic dynamics are represented in our benchmarks: the continuous control domains have deterministic dynamics, while the Gym Atari RAM games have stochastic dynamics. (In the Atari games, actions are repeated for a random number of frames.)

We use Trust Region Policy Optimization (TRPO) [20], a state-of-the-art policy gradient method, as our base reinforcement learning algorithm throughout our experiments, and we use the rllab implementations of TRPO and the continuous control tasks [5]. Full details for the experimental set-up are included in the appendix.

On all tasks, we compare against TRPO without intrinsic rewards, which we refer to as using naive exploration (in contrast to intrinsically motivated exploration). For the continuous control tasks, we also compare against intrinsic motivation using the $L_2$ model prediction error,

$$r_+(s, a, s') = \|s' - \mu_\phi(s, a)\|_2, \tag{12}$$

where $\mu_\phi$ is the mean of the learned Gaussian distribution $P_\phi$. The model prediction error was investigated as intrinsic motivation for deep reinforcement learning by Stadie et al [22], although they used a different method for learning the model $\mu_\phi$. This comparison helps us verify whether or not our proposed form of surprise, as a KL-divergence from the true dynamics model, is useful. Additionally, we compare our performance against the performance reported by Houthooft et al. [7] for Variational Information Maximizing Exploration (VIME), a method where the intrinsic reward associated with a transition approximates its Bayesian surprise using variational methods. Currently, VIME has achieved state-of-the-art results on intrinsic motivation for continuous control.

As a final check for the continuous control tasks, we benchmark the tasks themselves, by measuring the performance of the surprisal bonus without any dynamics learning: $r_+(s, a, s') = -\log P_{\phi_0}(s'|s, a)$, where $\phi_0$ are the original random parameters of $P_\phi$. This allows us to verify whether our benchmark tasks actually require surprise to solve at all, or if random exploration strategies successfully solve them.

### 4.1 CONTINUOUS CONTROL RESULTS

Median performance curves are shown in Figure 1 with interquartile ranges shown in shaded areas. Note that TRPO without intrinsic motivation failed on all tasks: the median score and upper quartile range for naive exploration were zero everywhere. Also note that TRPO with random exploration bonuses failed on most tasks, as shown separately in Figure 2. We found that surprise was not needed to solve MountainCar, but was necessary to perform well on the other tasks.

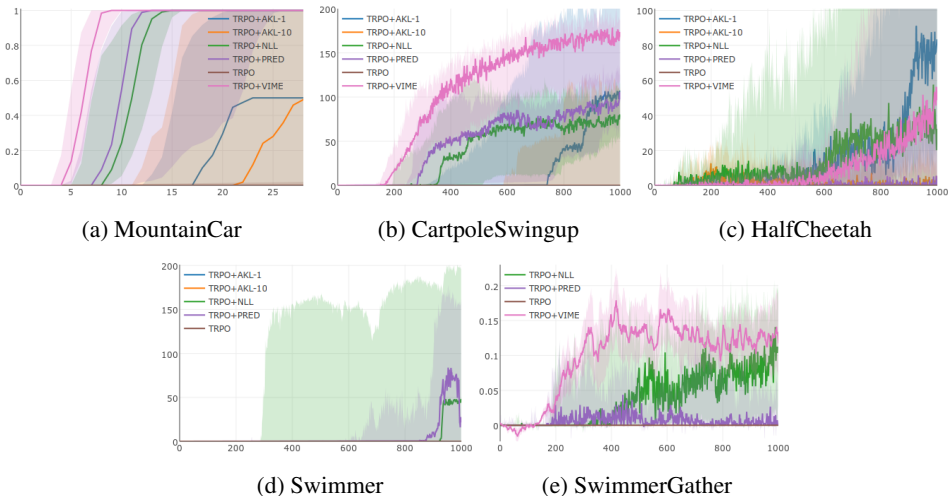

(a) MountainCar (b) CartpoleSwingup (c) HalfCheetah

(d) Swimmer (e) SwimmerGather

Figure 1: Median performance for the continuous control tasks over 10 runs with a fixed set of seeds, with interquartile ranges shown in shaded areas. The $x$-axis is iterations of training; the $y$-axis is average undiscounted return. AKL-$k$ refers to learning progress (8), NLL to surprisal (5), and PRED to (12). For the first four tasks, $\eta_0 = 0.001$; for SwimmerGather, $\eta_0 = 0.0001$. Results for VIME are from Houthooft et al. [7], reproduced here with permission. We note that the performance curve for VIME in the SwimmerGather environment represents only 2 random seeds, not 10.

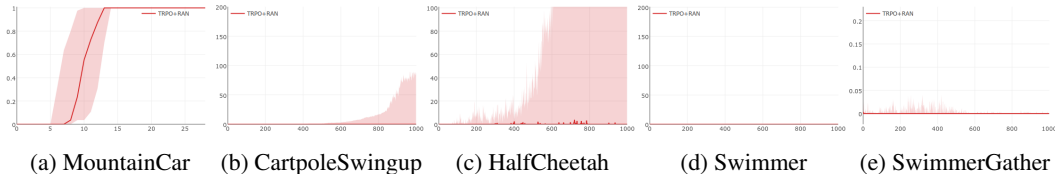

(a) MountainCar (b) CartpoleSwingup (c) HalfCheetah (d) Swimmer (e) SwimmerGather

Figure 2: Benchmarking the benchmarks: median performance for the continuous control tasks over 10 runs with a fixed set of seeds, with interquartile ranges shown in shaded areas, using the surprisal without learning bonus. RAN refers to the fact that this is essentially a random exploration bonus.

The surprisal bonus was especially robust across tasks, achieving good results in all domains and substantially exceeding the other baselines on the more challenging ones. The learning progress bonus for $k = 1$ was successful on CartpoleSwingup and HalfCheetah but it faltered in the others. Its weak performance in MountainCar was due to premature convergence of the dynamics model, which resulted in the agent receiving intrinsic rewards that were identically zero. (Given the simplicity of the environment, it is not surprising that the dynamics model converged so quickly.) In Swimmer, however, it seems that the learning progress bonuses did not inspire sufficient exploration. Because the Swimmer environment is effectively a stepping stone to the harder SwimmerGather, where the agent has to learn a motion primitive *and* collect target pellets, on SwimmerGather, we only evaluated the intrinsic rewards that had been successful on Swimmer.

Both surprisal and learning progress (with $k = 1$) exceeded the reported performance of VIME on HalfCheetah by learning to solve the task more quickly. On CartpoleSwingup, however, both were more susceptible to getting stuck in locally optimal policies, resulting in lower median scores than VIME. Surprisal performed comparably to VIME on SwimmerGather, the hardest task in the slate—in the sense that after 1000 iterations, they both reached approximately the same median score—although with greater variance than VIME.

Our results suggest that surprisal is a viable alternative to VIME in terms of performance, and is highly favorable in terms of computational cost. In VIME, a backwards pass through the dynamics model must be computed for every transition tuple separately to compute the intrinsic rewards, whereas our surprisal bonus only requires forward passes through the dynamics model for intrinsic

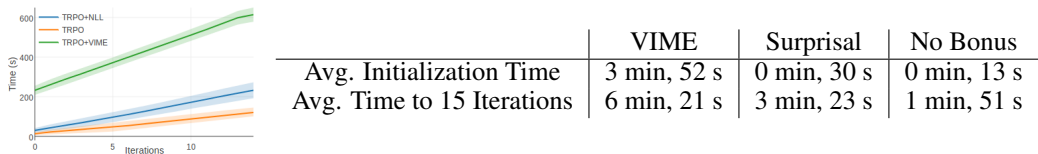

| | VIME | Surprisal | No Bonus |
|---|---|---|---|
| Avg. Initialization Time | 3 min, 52 s | 0 min, 30 s | 0 min, 13 s |
| Avg. Time to 15 Iterations | 6 min, 21 s | 3 min, 23 s | 1 min, 51 s |

Figure 3: Speed test: comparing the performance of VIME against our proposed intrinsic reward schemes, average compute time over 5 random runs. Tests were run on a Thinkpad T440p with four physical Intel i7-4700MQ cores, in the sparse HalfCheetah environment. VIME's greater initialization time, which is primarily spent in computation graph compilation, reflects the complexity of the Bayesian neural network model.

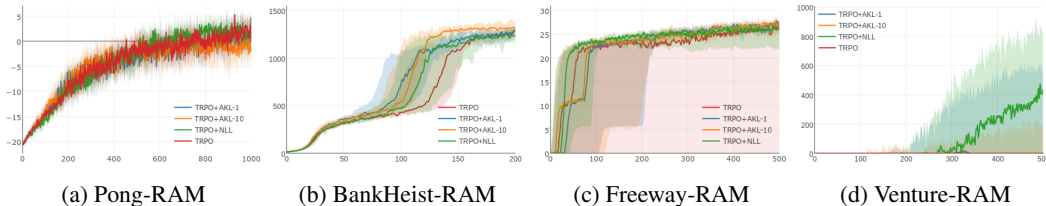

| (a) Pong-RAM | (b) BankHeist-RAM | (c) Freeway-RAM | (d) Venture-RAM |
|---|---|---|---|

Figure 4: Median performance for the Atari RAM tasks over 10 runs with a fixed set of seeds, with interquartile ranges shown in shaded areas. The $x$-axis is iterations of training; the $y$-axis is average undiscounted return. AKL-$k$ refers to learning progress (8), and NLL to surprisal (5).

reward computation. (Limitations of current deep learning tool kits make it difficult to efficiently compute separate backwards passes, whereas almost all of them support highly parallel forward computations.) Furthermore, our dynamics model is substantially simpler than the Bayesian neural network dynamics model of VIME. To illustrate this point, in Figure 3 we show the results of a speed comparison making use of the open-source VIME code [6], with the settings described in the VIME paper. In our speed test, our bonus had a per-iteration speedup of a factor of 3 over VIME.[2] We give a full analysis of the potential speedup in Appendix C.

## 4.2 ATARI RAM DOMAIN RESULTS

Median performance curves are shown in Figure 4, with tasks arranged from (a) to (d) roughly in order of increasing difficulty.

In Pong, naive exploration naturally succeeds, so we are not surprised to see that intrinsic motivation does not improve performance. However, this serves as a sanity check to verify that our intrinsic rewards do not degrade performance. (As an aside, we note that the performance here falls short of the standard score of 20 for this domain because we truncate play at 5000 timesteps.)

In BankHeist, we find that intrinsic motivation accelerates the learning significantly. The agents with surprisal incentives reached high levels of performance (scores > 1000) 10% sooner than naive exploration, while agents with learning progress incentives reached high levels almost 20% sooner.

In Freeway, the median performance for TRPO without intrinsic motivation was adequate, but the lower quartile range was quite poor—only 6 out of 10 runs ever found rewards. With the learning progress incentives, 8 out of 10 runs found rewards; with the surprisal incentive, all 10 did. Freeway is a game with very sparse rewards, where the agent effectively has to cross a long hallway before it can score a point, so naive exploration tends to exhibit random walk behavior and only rarely reaches the reward state. The intrinsic motivation helps the agent explore more purposefully.

---

[2]We compute this by comparing the marginal time cost incurred just by the bonus in each case: that is, if $T_{vime}$, $T_{surprisal}$, and $T_{nobonus}$ denote the times to 15 iterations, we obtain the speedup as

$$\frac{T_{vime} - T_{nobonus}}{T_{surprisal} - T_{nobonus}}.$$

In Venture, we obtain our strongest results in the Atari domain. Venture is extremely difficult because the agent has to navigate a large map to find very sparse rewards, and the agent can be killed by enemies interspersed throughout. We found that our intrinsic rewards were able to substantially improve performance over naive exploration in this challenging environment. Here, the best performance was again obtained by the surprisal incentive, which usually inspired the agent to reach scores greater than $500$.

### 4.3 COMPARING INCENTIVES

Among our proposed incentives, we found that surprisal worked the best overall, achieving the most consistent performance across tasks. The learning progress-based incentives worked well on some domains, but generally not as well as surprisal. Interestingly, learning progress with $k = 10$ performed much worse on the continuous control tasks than with $k = 1$, but we observed virtually no difference in their performance on the Atari games; it is unclear why this should be the case.

Surprisal strongly outperformed the $L_2$ error based incentive on the harder continuous control tasks, learning to solve them more quickly and without forgetting. Because we used fully-factored Gaussians for all of our dyanmics models, the surprisal had the form

$$-\log P_\phi(s'|s,a) = \sum_{i=1}^{n} \left( \frac{(s'_i - \mu_{\phi,i}(s,a))^2}{2\sigma^2_{\phi,i}(s,a)} + \log \sigma_{\phi,i}(s,a) \right) + \frac{k}{2}\log 2\pi,$$

which essentially includes the $L_2$-squared error norm as a sub-expression. The relative difference in performance suggests that the variance terms confer additional useful information about the novelty of a state-action pair.

## 5 RELATED WORK

Substantial theoretical work has been done on optimal exploration in finite MDPs, resulting in algorithms such as $E^3$ [10], R-max [3], and UCRL [9], which scale polynomially with MDP size. However, these works do not permit obvious generalizations to MDPs with continuous state and action spaces. C-PACE [18] provides a theoretical foundation for PAC-optimal exploration in MDPs with continuous state spaces, but it requires a metric on state spaces. Lopes et al. [11] investigated exploration driven by learning progress and proved theoretical guarantees for their approach in the finite MDP case, but they did not address the question of scaling their approach to continuous or high-dimensional MDPs. Also, although they formulated learning progress in the same way as (8), they formed intrinsic rewards differently. Conceptually and mathematically, our work is closest to prior work on curiosity and surprise [8, 19, 23, 24], although these works focus mainly on small finite MDPs.

Recently, several intrinsic motivation strategies that deal specifically with deep reinforcement learning have been proposed. Stadie et al. [22] learn deterministic dynamics models by minimizing Euclidean loss—whereas in our work, we learn stochastic dynamics with cross entropy loss—and use $L_2$ prediction errors for intrinsic motivation. Houthooft et al. [7] train Bayesian neural networks to approximate posterior distributions over dynamics models given observed data, by maximizing a variational lower bound; they then use second-order approximations of the Bayesian surprise as intrinsic motivation. Bellemare et al. [2] derived pseudo-counts from CTS density models over states and used those to form intrinsic rewards, notably resulting in dramatic performance improvement on Montezuma's Revenge, one of the hardest games in the Atari domain. Mohamed and Rezende [14] developed a scalable method of approximating empowerment, the mutual information between an agent's actions and the future state of the environment, using variational methods. Oh et al. [16] estimated state visit frequency using Gaussian kernels to compare against a replay memory, and used these estimates for directed exploration.

## 6 CONCLUSIONS

In this work, we formulated surprise for intrinsic motivation as the KL-divergence of the true transition probabilities from learned model probabilities, and derived two approximations—surprisal and $k$-step

learning progress—that are scalable, computationally inexpensive, and suitable for application to high-dimensional and continuous control tasks. We showed that empirically, motivation by surprisal and 1-step learning progress resulted in efficient exploration on several hard deep reinforcement learning benchmarks. In particular, we found that surprisal was a robust and effective intrinsic motivator, outperforming other heuristics on a wide range of tasks, and competitive with the current state-of-the-art for intrinsic motivation in continuous control.

## ACKNOWLEDGEMENTS

We thank Rein Houthooft for interesting discussions and for sharing data from the original VIME experiments. We also thank Rocky Duan, Carlos Florensa, Vicenc Rubies-Royo, Dexter Scobee, and Eric Mazumdar for insightful discussions and reviews of the preliminary manuscript.

This work is supported by TRUST (Team for Research in Ubiquitous Secure Technology) which receives support from NSF (award number CCF-0424422).

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

## A    SINGLE STEP SECOND-ORDER OPTIMIZATION

In our experiments, we approximately solve several optimization problems by using a single second-order step with a line search. This section will describe the exact methodology, which was originally given by Schulman et al. [20].

We consider the optimization problem

$$p^* = \max_{\theta} L(\theta) \ : \ D(\theta) \leq \delta, \tag{13}$$

where $\theta \in R^n$, and for some $\theta_{old}$ we have $D(\theta_{old}) = 0$, $\nabla_\theta D(\theta_{old}) = 0$, and $\nabla_\theta^2 D(\theta_{old}) \succeq 0$; also, $\forall \theta, D(\theta) \geq 0$.

We suppose that $\delta$ is small, so the optimal point will be close to $\theta_{old}$. We also suppose that the curvature of the constraint is much greater than the curvature of the objective. As a result, we feel justified in approximating the objective to linear order and the constraint to quadratic order:

$$L(\theta) \approx L(\theta_{old}) + g^T(\theta - \theta_{old}) \qquad g \doteq \nabla_\theta L(\theta_{old})$$

$$D(\theta) \approx \frac{1}{2}(\theta - \theta_{old})^T A(\theta - \theta_{old}) \qquad A \doteq \nabla_\theta^2 D(\theta_{old}).$$

We now consider the approximate optimization problem,

$$p^* \approx \max_\theta g^T(\theta - \theta_{old}) \; : \; \frac{1}{2}(\theta - \theta_{old})^T A(\theta - \theta_{old}) \leq \delta.$$

This optimization problem is convex as long as $A \succeq 0$, which is an assumption that we make. (If this assumption seems to be empirically invalid, then we repair the issue by using the substitution $A \rightarrow A + \epsilon I$, where $I$ is the identity matrix, and $\epsilon > 0$ is a small constant chosen so that we usually have $A + \epsilon I \succeq 0$.) This problem can be solved analytically by applying methods of duality, and its optimal point is

$$\theta^* = \theta_{old} + \sqrt{\frac{2\delta}{g^T A^{-1} g}} A^{-1} g. \tag{14}$$

It is possible that the parameter update step given by (14) may not exactly solve the original optimization problem (13)—in fact, it may not even satisfy the constraint—so we perform a line search between $\theta_{old}$ and $\theta^*$. Our update with the line search included is given by

$$\theta = \theta_{old} + s^k \sqrt{\frac{2\delta}{g^T A^{-1} g}} A^{-1} g, \tag{15}$$

where $s \in (0,1)$ is a backtracking coefficient, and $k$ is the smallest integer for which $L(\theta) \geq L(\theta_{old})$ and $D(\theta) \leq \delta$. We select $k$ by checking each of $k = 1, 2, ..., K$, where $K$ is the maximum number of backtracks. If there is no value of $k$ in that range which satisfies the conditions, no update is performed.

Because the optimization problems we solve with this method tend to involve thousands of parameters, inverting $A$ is prohibitively computationally expensive. Thus in the implementation of this algorithm that we use, the search direction $x = A^{-1} g$ is found by using the conjugate gradient method to solve $Ax = g$; this avoids the need to invert $A$.

When $A$ and $g$ are sample averages meant to stand in for expectations, we employ an additional trick to reduce the total number of computations necessary to solve $Ax = g$. The computation of $A$ is more expensive than $g$, and so we use a smaller fraction of the population to estimate it quickly. Concretely, suppose that the original optimization problem's objective is $E_{z \sim P}[L(\theta, z)]$, and the constraint is $E_{z \sim P}[D(\theta, z)] \leq \delta$, where $z$ is some random variable and $P$ is its distribution; furthermore, suppose that we have a dataset of samples $D = \{z_i\}_{i=1,...,N}$ drawn on $P$, and we form an approximate optimization problem using these samples. Defining $g(z) \doteq \nabla_\theta L(\theta_{old}, z)$ and $A(z) \doteq \nabla_\theta^2 D(\theta_{old}, z)$, we would need to solve

$$\left( \frac{1}{|D|} \sum_{z \in D} A(z) \right) x = \frac{1}{|D|} \sum_{z \in D} g(z)$$

to obtain the search direction $x$. However, because the computation of the average Hessian is expensive, we sub-sample a batch $b \subset D$ to form it. As long as $b$ is a large enough set, then the approximation

$$\frac{1}{|b|} \sum_{z \in b} A(z) \approx \frac{1}{|D|} \sum_{z \in D} A(z) \approx \underset{z \sim P}{E}[A(z)]$$

is good, and the search direction we obtain by solving

$$\left( \frac{1}{|b|} \sum_{z \in b} A(z) \right) x = \frac{1}{|D|} \sum_{z \in D} g(z)$$

is reasonable. The sub-sample ratio $|b|/|D|$ is a hyperparameter of the algorithm.

## B  EXPERIMENT DETAILS

### B.1  ENVIRONMENTS

The environments have the following state and action spaces: for the sparse MountainCar environment, $S \subseteq \mathbb{R}^2$, $A \subseteq \mathbb{R}^1$; for the sparse CartpoleSwingup task, $S \subseteq \mathbb{R}^4$, $A \subseteq \mathbb{R}^1$; for the sparse HalfCheetah

task, $S \subset \mathbb{R}^{20}, A \subseteq \mathbb{R}^6$; for the sparse Swimmer task, $S \subseteq \mathbb{R}^{13}, A \subseteq \mathbb{R}^2$; for the SwimmerGather task, $S \subseteq \mathbb{R}^{33}, A \subseteq \mathbb{R}^2$; for the Atari RAM domain, $S \subseteq \mathbb{R}^{128}, A \subseteq \{1, ..., 18\}$.

For the sparse MountainCar task, the agent receives a reward of $1$ only when it escapes the valley. For the sparse CartpoleSwingup task, the agent receives a reward of $1$ only when $\cos(\beta) > 0.8$, with $\beta$ the pole angle. For the sparse HalfCheetah task, the agent receives a reward of $1$ when $x_{body} \geq 5$. For the sparse Swimmer task, the agent receives a reward of $1 + |v_{body}|$ when $|x_{body}| \geq 2$.

Atari RAM states, by default, take on values from $0$ to $256$ in integer intervals. We use a simple preprocessing step to map them onto values in $(-1/3, 1/3)$. Let $x$ denote the raw RAM state, and $s$ the preprocessed RAM state:

$$s = \frac{1}{3}\left(\frac{x}{128} - 1\right).$$

## B.2    Policy and Value Functions

For all continuous control tasks we used fully-factored Gaussian policies, where the means of the action distributions were the outputs of neural networks, and the variances were separate trainable parameters. For the sparse MountainCar and sparse CartpoleSwingup tasks, the policy mean networks had a single hidden layer of 32 units. For sparse HalfCheetah, sparse Swimmer, and SwimmerGather, the policy mean networks were of size $(64, 32)$. For the Atari RAM tasks, we used categorical distributions over actions, produced by neural networks of size $(64, 32)$.

The value functions used for the sparse MountainCar and sparse CartpoleSwingup tasks were neural networks with a single hidden layer of 32 units. For sparse HalfCheetah, sparse Swimmer, and SwimmerGather, time-varying linear value functions were used, as described by Duan et al. [5]. For the Atari RAM tasks, the value functions were neural networks of size $(64, 32)$. The neural network value functions were learned via single second-order step optimization; the linear baselines were obtained by least-squares fit at each iteration.

All neural networks were feed-forward, fully-connected networks with $\tanh$ activation units.

## B.3    TRPO Hyperparameters

For all tasks, the MDP discount factor $\gamma$ was fixed to $0.995$, and generalized advantage estimators (GAE) [21] were used, with the GAE $\lambda$ parameter fixed to $0.95$.

In the table below, we show several other TRPO hyperparameters. Batch size refers to steps of experience collected at each iteration. The sub-sample factor is for the second-order optimization step, as detailed in Appendix A.

| Environments | Batch Size | Sub-Sample | Max Rollout Length | $\delta_{KL}$ |
|---|---|---|---|---|
| Mountaincar, Cartpole Swingup | 5000 | 1 | 500 | 0.01 |
| HalfCheetah, Swimmer | 5000 | 1 | 500 | 0.05 |
| SwimmerGather | 50,000 | 0.1 | 500 | 0.01 |
| Pong | 10,000 | 1 | 5000 | 0.01 |
| Bankheist, Freeway | 13,500 | 1 | 5000 | 0.01 |
| Venture | 50,000 | 0.2 | 7000 | 0.01 |

Table 1: TRPO hyperparameters for our experiments.

## B.4    Exploration Hyperparameters

For all tasks, fully-factored Gaussian distributions were used as dynamics models, where the means and variances of the distributions were the outputs of neural networks.

For the sparse MountainCar and sparse CartpoleSwingup tasks, the means and variances were parametrized by single hidden layer neural networks with 32 units. For all other tasks, the means and variances were parametrized by neural networks with two hidden layers of size 64 units each. All networks used $\tanh$ activation functions.

For all continuous control tasks except SwimmerGather, we used replay memories of size $5,000,000$, and a KL-divergence step size of $\kappa = 0.001$. For SwimmerGather, the replay memory was the same size, but we set the KL-divergence size to $\kappa = 0.005$. For the Atari RAM domain tasks, we used replay memories of size $1,000,000$, and a KL-divergence step size of $\kappa = 0.01$.

For all tasks except SwimmerGather and Venture, 5000 time steps of experience were sampled from the replay memory at each iteration of dynamics model learning to take a stochastic step on (11), and a sub-sample factor of 1 was used in the second-order step optimizer. For SwimmerGather and Venture, $10,000$ time steps of experience were sampled at each iteration, and a sub-sample factor of $0.5$ was used in the optimizer.

For all continuous control tasks, the $L_2$ penalty coefficient was set to $\alpha = 1$. For the Atari RAM tasks except for Venture, it was set to $\alpha = 0.01$. For Venture, it was set to $\alpha = 0.1$.

For all continuous control tasks except SwimmerGather, $\eta_0 = 0.001$. For SwimmerGather, $\eta_0 = 0.0001$. For the Atari RAM tasks, $\eta_0 = 0.005$.

## C  ANALYSIS OF SPEEDUP COMPARED TO VIME

In this section, we provide an analysis of the time cost incurred by using VIME or our bonuses, and derive the potential magnitude of speedup attained by our bonuses versus VIME.

At each iteration, bonuses based on learned dynamics models incur two primary costs:

- the time cost of fitting the dynamics model,
- and the time cost of computing the rewards.

We denote the dynamics fitting costs for VIME and our methods as $T_{vime}^{fit}$ and $T_{ours}^{fit}$. Although the Bayesian neural network dynamics model for VIME is more complex than our model, the fit times can work out to be similar depending on the choice of fitting algorithm. In our speed test, the fit times were nearly equivalent, but used different algorithms.

For the time cost of computing rewards, we first introduce the following quantities:

- $n$: the number of CPU threads available,
- $t_f$: time for a forward pass through the model,
- $t_b$: time for a backward pass through the model,
- $N$: batch size (number of samples per iteration),
- $k$: the number of forward passes that can be performed simultaneously.

For our method, the time cost of computing rewards is

$$T_{ours}^{rew} = \frac{N t_f}{kn}.$$

For VIME, things are more complex. Each reward requires the computation of a *gradient* through its model, which necessitates a forward and a backward pass. Because gradient calculations cannot be efficiently parallelized by any deep learning toolkits currently available[3], each $(s, a, s')$ tuple requires its own forward/backward pass. As a result, the time cost of computing rewards for VIME is:

$$T_{vime}^{rew} = \frac{N(t_f + t_b)}{n}.$$

The speedup of our method over VIME is therefore

$$\frac{T_{vime}^{fit} + \frac{N(t_f + t_b)}{n}}{T_{ours}^{fit} + \frac{N t_f}{kn}}.$$

In the limit of large $N$, and with the approximation that $t_f \approx t_b$, the speedup is a factor of $\sim 2k$.

---

[3]If this is not correct, please contact the authors so that we can issue a correction! But to the best of our knowledge, this is currently true, at time of publication.

