# Peer review of "Surprise-Based Intrinsic Motivation for Deep Reinforcement Learning"

_ICLR 2017 — rejected_

[Official Review · AnonReviewer2 · rating 6 · confidence 4 · 16 Dec 2016]
**No Title**

This paper explores the topic of intrinsic motivation in the context of deep RL. It proposes a couple of variants derived from an auxiliary model-learning process (prediction error, surprise and learning progress), and shows that those can help exploration on a number of continuous control tasks (and the Atari game “venture”, maybe).

Novelty: none of the proposed types of intrinsic motivation are novel, and it’s arguable whether the application to deep RL is novel (see e.g. Kompella et al 2012).

Potential: the idea of seeking out states where a transition model is uncertain is sensible, but also limited -- I would encourage the authors to also discuss the limitations. For example in a game like Go the transition model is trivially learned, so this approach would revert to random exploration. So other forms of learning progress or surprise derived from the agent’s competence instead might be more promising in the long run? See also Srivastava et al 2012 for further thoughts.

Computation time: I find the paper’s claimed superiority over VIME to be overblown: the gain seems to stem almost exclusively from a faster initialization, but have very similar per-step cost? So given that VIME is also performing very competitively, what arguments can you advance for your own method(s)?

[Official Review · AnonReviewer3 · rating 6 · confidence 3 · 17 Dec 2016]

This paper provides a surprise-based intrinsic reward method for reinforcement learning, along with two practical algorithms for estimating those rewards. The ideas are similar to previous work in intrinsic motivation (including VIME and other work in intrinsic motivation). 
As a positive, the methods are simple to implement, and provide benefits on a number of tasks.
However, they are almost always outmatched by VIME, and not one of their proposed method is consistently the best of those proposed (perhaps the most consistent is the surprisal, which is unfortunately not asymptotically equal to the true reward). The authors claim massive speed up, but the numerical measurements show that VIME is slower to initialize but not significantly slower per iteration otherwise (perhaps a big O analysis would clarify the claims).
Overall it's a decent, simple technique, perhaps slightly incremental on previous state of the art.

[Official Review · AnonReviewer1 · rating 6 · confidence 3 · 18 Dec 2016]
**Important extension of existing work on intrinsic motivation. Experimental results are less convincing.**

The authors present a novel approach to surprise-based intrinsic motivation in deep reinforcement learning. The authors clearly explain the difference from other recent approaches to intrinsic motivation and back up their method with results from a broad class of discrete and continuous action domains. They present two tractable approximations to their framework - one which ignores the stochasticity of the true environmental dynamics, and one which approximates the rate of information gain (somewhat similar to Schmidhuber's formal theory of creativity, fun and intrinsic motivation). The results of this exploration bonus when added to TRPO are generally better than standard TRPO. However, I would have appreciated a more thorough comparison against other recent work on intrinsic motivation. For instance, Bellemare et al 2016 recently achieved significant performance gains on challenging Atari games like Montezuma's Revenge by combining DQN with an exploration bonus, however Montezuma's Revenge is not presented as an experiment here. Such comparisons would significantly improve the strength of the paper.

[Final Decision · Program Chairs · 06 Feb 2017]
**ICLR committee final decision**

The paper proposes an intuitive method for exploration, namely to build a model of the system dynamics and explore regions where this approximation differs from the observed data (i.e., how "surprised" the agent was by an observation). The idea is a nice one, and part of the benefit comes from the simplicity and wide applicability of the approach.
 
 The main drawback of this paper is simply that the resulting approach doesn't substantially outperform existing approaches, at least not to a degree where it seem like the paper should should be clearly accepted to ICLR. On the continuous control tasks, the advantage over VIME seems very unclear (at best the results are mixed, showing sometimes surprisal and sometime VIME do better), and on the Atari games no comparison is made against many of the methods tuned for this setting, such as Gorila (Nair, 2015) which achieves some of the best results we are aware of on the Venture game, which is definitely the strongest result in this current paper, but not as good as this previous work. We know the settings are different, but overall it seems like the approach is largely outperformed by existing approaches, and thus the advantage mainly comes from runtime. This is certainly an interesting take, but needs to be studied a lot more thoroughly before it would make a really compelling case. We would like to recomend this paper to the workshop track. The pros/cons are as follows:
 
 Pros:
 + Simple and intuitive method for exploration
 
 Cons:
 - Doesn't seem to substantially outperform existing methods
 - No comparison to many alternative approaches for some of the "better" results in the paper.